# Characterization of Nano-Scale Parallel Lamellar Defects in RDX and HMX Single Crystals by Two-Dimension Small Angle X-ray Scattering

**DOI:** 10.3390/molecules27123871

**Published:** 2022-06-16

**Authors:** Haobin Zhang, Jinjiang Xu, Shichun Li, Jie Sun, Xiaolin Wang

**Affiliations:** 1Institute of Nuclear Physics and Chemistry, China Academy of Engineering Physics, Mianyang 621900, China; zhhb03@caep.cn; 2Institute of Chemical Materials, China Academy of Engineering Physics, Mianyang 621900, China; xujinjiang@caep.cn (J.X.); lishichun@caep.cn (S.L.); sunjie@caep.cn (J.S.); 3China Academy of Engineering Physics, Mianyang 621900, China

**Keywords:** nano-scale defects, single crystal, small angle X-ray scattering, explosive, binding energy

## Abstract

Nano-scale crystal defects extremely affect the security and reliability of explosive charges of weapons. In this work, the nano-scale crystal defects of 1,3,5-trinitro-1,3,5-triazacyclohexane (RDX) and octahydro-1,3,5,7-tetranitro-1,3,5,7-tetrazocine (HMX) single crystals were characterized by two-dimension SAXS. Deducing from the changes of SAXS pattern with sample stage rotating, we firstly found the parallel lamellar nano-scale defects in both RDX and HMX single crystals. Further analysis shows that the average diameter and thickness of nano-scale lamellar defects for RDX single crystal are 66.4 nm and 19.3 nm, respectively. The results of X-ray diffraction (XRD) indicate that the lamellar nano-scale defects distribute along the (001) in RDX and the (011) in HMX, which are verified to be the crystal planes with the lowest binding energy by the theoretical calculation.

## 1. Introduction

Nano-scale crystal defects extremely affect the security and reliability of explosive charges of weapons [1,2,3,4,5,6,7]. On the one hand, defects play a key role in the initiation of explosives resulting from mechanical stimuli, such as shock loading. Plastic deformation gives rise to numerous energy localization mechanisms, which can lead to the formation of hot spots [8,9]. On the other hand, defects can decrease the strength of explosive charges, and the evolution of various defect structures often leads to cracking of explosive charges in the process of storage and usage [10,11]. Accurate characterization of crystal defects is of great importance to the application of explosives.

The defects of explosive crystals at a micron scale are relatively easy to be characterized by Computed Tomography (CT) [12], Optical Microscopes (OM) [13], SEM [14], and plane-polarized light microscopy [15]. Nano-scale defect is the microcosmic origin of damage evolution and macroscopic cracking of explosive charges. However, the nano-scale defects of explosive crystals are difficult to be detected because the defects are distributed in the explosive crystal with a large number but small scale, and the explosive material is easy to decompose under the high-energy stimulation from characterization methods.

To characterize the nano-scale defects, small angle X-ray/neutron scattering technology (SAXS/SANS) is a relatively good choice. By using SAXS and SANS to test the radius of defects in explosives, assuming that the shape of defects is spherical or cylindrical morphology, Stepanov et al. studied the distribution of nano-scale voids of three kinds of RDX-based PBXs by USAXS [16], Trevor et al. studied the relationship between nano-scale defects number and their irreversible expansion for TATB-based PBXs during thermal cycles [17], Yan et al. analyzed the size changes of the nano-scale defects in octahydro-1,3,5,7-tetranitro-1,3,5,7-tetrazocine (HMX)-based PBXs [18,19], and Mang et al. distinguished the crystal defects of explosives by changing pressure [20,21,22]. Peterson et al. quantitatively analyzed the damage in an HMX-based composite explosive subjected to a linear thermal gradient by means of SAXS and OM [23,24], the SAXS data explicitly demonstrate an increase in the crack density and the formation of a new population of voids with increasing temperature.

However, SAXS usually measures the total porosity of a system. In the studies on explosive powders and pressed charges, the measured porosity is a combination of inter- and intra-granular voids (defects) [25,26,27]. This fact complicates SAXS analysis and often limits morphological interpretation. Without contrast variation, there is no way to discern between the two types of voids, which hinders the study of the process and mechanism of defect evolution. By studying the crystal defects using single crystal, we can only obtain the defects inside the particles without disturbing from inter-granular defects.

In this paper, HMX and 1,3,5-trinitro-1,3,5-triazacyclohexane (RDX) single crystals were used to characterize the shape of nano-scale defects for the first time by 2D SAXS. 3D shapes of nano-scale defects can be obtained by rotating the sample stage. The X-ray diffraction (XRD) measurements are used to characterize the distributing orientation of the nano-scale defects, and a theory computation by the Forcite module in the Material Studio was used to explain the formation mechanism of nano-scale defects.

## 2. Materials and Methods

### 2.1. Preparation of Single Crystals

RDX and HMX particles were provided by the Institute of Chemical Materials, China Academy of Engineering Physics (CAEP). Anhydrous acetone (AR) was purchased from Chengdu Kelong Chemical Reagent Factory. 10 g of RDX was dissolved in 100 mL of acetone to make a saturated solution, then an additional 20 mL of acetone was added to fully dissolve it. Acetone solvent was slowly volatilized at a constant temperature to obtain a large RDX single crystal, which was ground into thin slices with 1 mm of thickness along the direction of the maximum exposed crystal surface, which was characterized to be the (120) crystal surface. Similar methods were used to obtain HMX single crystal with 1 mm thickness along the (001) crystal surface.

### 2.2. Characterization Methods

The SAXS measurements were carried out on a Xeuss 2.0 system of Xencos France equipped with a multi-layer focused Cu Kα X-ray source (GenilX3D Cu ULD, Xencos SA, France), operating at a maximum power of 30 W. The wavelength of the X-ray radiation was 0.154 nm. Two pairs of scatterless slits were located 1500 mm apart from each other for collimating the X-ray beam. Scattering data were recorded with the aid of a Pilatus 300 K detector (DECTRIS, Swiss, resolution: 487 × 640, pixel size = 172 µm). The sample-to-detector distance was 2486 mm. Scattering curves were obtained in the q-range, q = 4π sin θ/λ, between 0.06 and 2.0 nm^−1^, q being the scattering vector, and 2θ the scattering angle. Standard measurement conditions were 60 kV, 0.5 mA, and 1800 s (acquisition time). Each SAXS pattern was background corrected and normalized using the standard procedure.

The XRD Powder X-ray diffraction experiments were performed using a Bruker D8 Advance X-ray diffractometer equipped with a Cu tube and Ni-filter, yielding Cu K_α1_ and Cu K_α2_ radiation (λ_1_ = 1.5406 Å, λ_2_ = 1.5444 Å, respectively). The diffractometer was operated in Bragg-Brentano geometry with the data collected using a Vantec linear position-sensitive detector (PSD). The X-ray tube operating conditions were 40 kV and 40 mA. Data collection was performed over the range of 2θ between 10 and 50° using a step size of Δ2θ = 0.01°and a counting time of 0.5 s/step.

### 2.3. Theoretic Calculation

Binding energy (ΔE) between crystal surface is a measure of the inter-planar action of crystal, where a low, positive value indicates a weak interaction between the crystal surfaces. We built periodic supercell (4 × 4 × 4) bulks of RDX and HMX and cleaved different crystal surface in the bulk RDX and HMX, finally, we rebuilt periodic supercell (4 × 1 × 4) with a vacuum thickness of 20 Å. The structure of RDX and HMX slabs is shown in Figure 1. The optimization of the RDX and HMX bulks and slabs was calculated by a classical force field simulation method in the Forcite module [28] with COMPASS force field [29]. The structures were relaxed with the following thresholds for the converged structure: energy change per atom was less than 1.0 × 10^−5^ eV, residual force was less than 0.03 eV/Å, the displacement of atoms during the geometry optimization was less than 0.001 Å, and the residual bulk stress less than 0.05 GPa. We calculate the Δ*E* in a slab model according to Formula (1):(1)$$\Delta E=\frac{E_{\mathrm{slab}}-E_{\mathrm{bulk}}}{S}$$
where *E*_slab_ and *E*_bulk_ are the total energies of the surface slab and bulk super-cell, respectively. *S* is the area of the surface slab.

## 3. Results and Discussion

### 3.1. Nano-Scale Defect Structure of RDX and HMX Single Crystal

The structure of nano-scale defects in explosive crystals was studied by Small Angle X-ray Scattering (SAXS) technique. Taking RDX and HMX single crystals as the research object, the quasi-three-dimensional SAXS test of RDX and HMX single crystals was realized by rotating the sample stage, as shown in Figure 2.

In the SAXS patterns of RDX, strong streak was found from the front side. It shows that the crystal defects in RDX single crystal are lamellar or rod-like with preferred orientation. To confirm the crystal defects shape, the SAXS patterns from different directions were obtained by rotating the sample stage. The streak did not disappear during the horizontal direction orientation but disappeared during the vertical direction orientation, which illustrates that the nano-scale defects are lamellar structure along the horizontal direction. The SAXS scattering signal shows obvious orientation, indicating the lamellar defects in RDX single crystal are parallel to each other. According to the results, the nano-sized defects in RDX single crystal are a lamellar structure distributing along the same direction, as schematized in Figure 3.

In the case of HMX single crystal, a similar phenomenon was observed. However, except for the main streak, there is a second streak in the SAXS patterns, which means there are some lamellar crystal defects directed in the other orientation.

To obtain the size of the orientated lamellar crystal defects, SAXS scattering data were processed. Since the scattering signal is clearly oriented, the integral can be carried out along the orientation direction (S_12_), as shown in Figure 4. For the lamellar defect, the scattering signal along the S_12_ direction satisfies the Guinier Formula [30]:(2)$$\ln I(q)=\ln I_{0}(q)-\frac{R_{g}{}^{2}q^{2}}{3}$$
where *q* is the scattering vector, *I*(*q*) is the scattering intensity, *I*_0_(*q*) is the scattering intensity that is extrapolated to zero, and *R*g is the rotation radius of the lamellar scattering body. According to Formula (2), ln *I*(*q*) was used to plot with *q*^2^, which formed a straight line in the low-angle direction for linear fitting, so that the average thickness of nano-scale lamellar defect in RDX single crystal was 19.3 nm.

Since the nano-scale defects in the single crystal of RDX explosives are lamellar defects parallel to each other, the length defining the preferred orientation of the crystal defects was calculated by Ruland’s streak method [31,32] with Gaussian distribution function:(3)$$B_{\mathrm{rad}}^{2}=L^{-2}s^{-2}+B_{\mathrm{eq}}^{2}$$
where *B*_rad_ is the half-peak width of the Azimuthal peak, L is the length of the preferred scattering body, *s* is the scattering vectors, and *B*_eq_ is the *B*_rad_ that is extrapolated to s-0.

By integrating SAXS data along the azimuth Angle (Figure 5a), the half-peak width Brad corresponding to different scattering vectors *s* was obtained, and *B*_rad_^2^ was used to plot s^−2^ (Figure 5b), so that the average diameter of the lamellar single crystal was 66.4 nm, while *B*_eq_ = 0 indicated that all the lamellar defects were completely parallel to each other.

### 3.2. Orientation of the Nano-Scale Lamellar Defects

Powder X-ray diffractometer (XRD) was used to test the orientation of the lamellar defect, as shown in Figure 6. According to Bragg’s law, the diffraction pattern of (002), (004), and (006) crystal plane of RDX is the different diffraction orders of (001) crystal plane, which is parallel to the plane where the lamellar defect lies. Similarly, the plane where the lamellar defect lies is crystal plane (011) in HMX single crystal.

### 3.3. Mechanism of the Nano-Scale Defects Orientation

In order to explain the reason why the lamellar defect is distributed along the specific crystal plane in RDX and HMX single crystals, the binding energies of different crystal planes are calculated by using the MD simulations. Forcite module in Material Studio 6 was used to construct the slab. After adding a specific vacuum layer in different directions, the energy difference with the complete crystal was calculated to obtain the minimum inter-crystal binding energy along different crystal planes, as listed in Table 1.

In RDX single crystal, the crystal plane (001) is of the lowest binding energy, while the XRD test results showed that the lamellar defect in RDX was distributed exactly along the crystal plane (001). This indicates that the reason for the formation of lamellar defects distributed along (001) crystal surface in RDX crystal is that the crystal surface with the least binding energy leads to the cracking along (001) direction.

In the HMX single crystal, the crystal plane (011) is of the lowest binding energy, consistent with the orientation of the lamellar defect in HMX. It can be concluded that the HMX single crystal possesses many nano-scale defects along the (011) crystal plane with the least binding energy. The formation of lamellar defects is influenced by the objective factor of crystal structure, so it is speculated that there are similar nano-scale defects in explosive powder. This kind of lamellar defect can provide important significance for further study of defect distribution and evolution.

## 4. Conclusions

In the present contribution, we have demonstrated many parallel lamellar nano-scale defects both in RDX and HMX single crystals by two-dimension SAXS. Further analysis shows that the average diameter and thickness of nano-scale lamellar defects for RDX single crystal are 66.4 nm and 19.3 nm, respectively. The results of XRD patterns indicate that the lamellar nano-scale defects distribute along the (001) crystal plane in RDX and the (011) in HMX. To demonstrate the mechanism of parallel lamellar nano-scale defects, the binding energy between crystal planes are calculated by the Material Studio. The lamellar nano-scale defects just distribute along the crystal planes with the lowest binding energy. We can deduce that the RDX and HMX single crystal can easily crack along the crystal plane with the least binding energy.

## Figures and Tables

**Figure 1 molecules-27-03871-f001:**
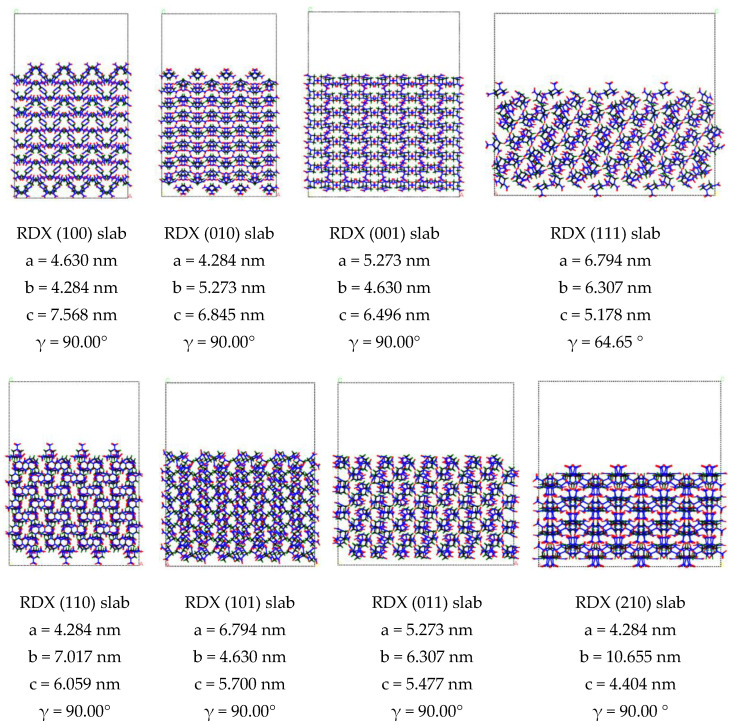
The slab of RDX (a) and HMX (b) crystal lattice constructed within Forcite module in Material Studio 6.

**Figure 2 molecules-27-03871-f002:**
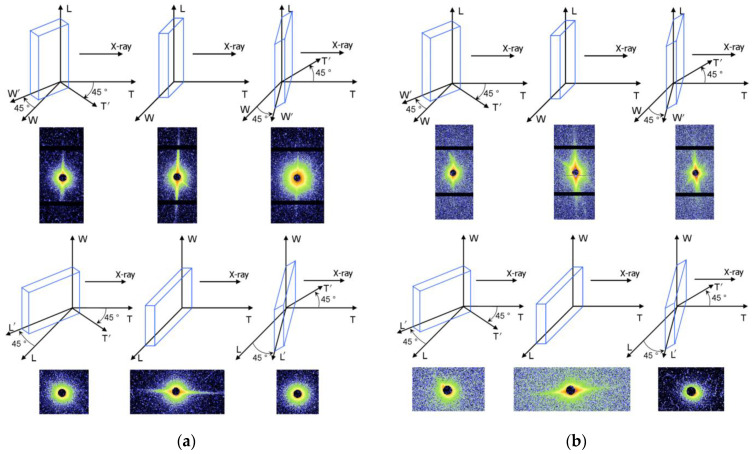
SAXS patterns of nano-scale crystal defects in explosive single crystal changing with sample stage rotating: (**a**) In RDX single crystal, (**b**) in HMX single crystal.

**Figure 3 molecules-27-03871-f003:**
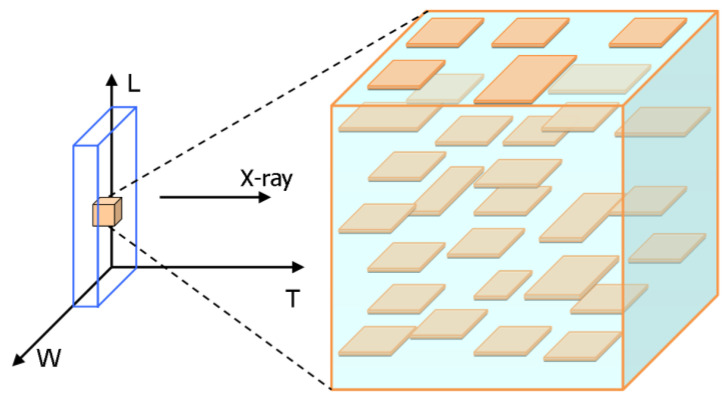
Schematic diagram of shape and distribution of nano-scale defects in RDX and HMX single crystal.

**Figure 4 molecules-27-03871-f004:**
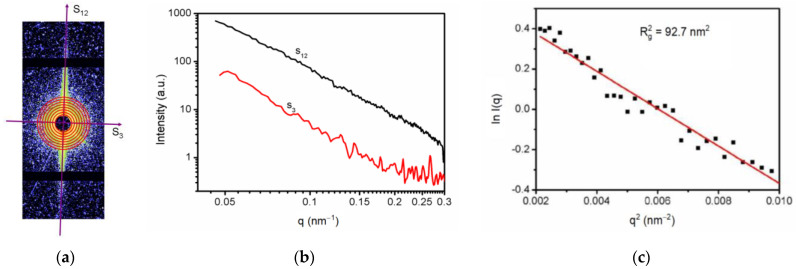
SAXS data processing: (**a**) Scattering direction punctuating, (**b**) SAXS curve of RDX single crystals along the scattering direction S_12_ and S_3_, (**c**) SAXS data analyzing by the Guinier fitting curve along the S_12_.

**Figure 5 molecules-27-03871-f005:**
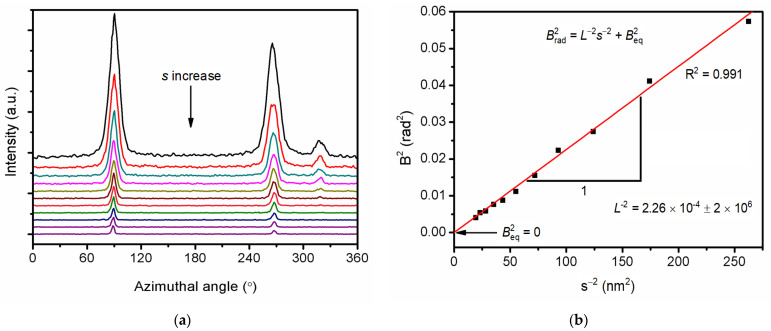
SAXS data analyzing by the Ruland method: (**a**) Azimuth integral of SAXS data, (**b**) Ruland fitting result.

**Figure 6 molecules-27-03871-f006:**
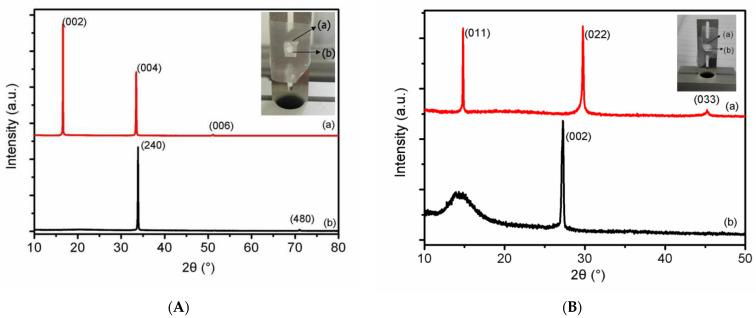
Determination of crystal plane directions of single crystals by the XRD: (**A**) Crystal plane of RDX single crystal, (**B**) crystal plane of HMX single crystal.

**Table 1 molecules-27-03871-t001:** The binding energies of RDX and HMX single crystals along different crystal surfaces.

Sample	Crystal Face	E (kcal/mol)	E (eV)	∆E (eV)	A (nm)	B (nm)	Θ (°)	S (nm^2^)	∆E/S (eV·nm^−2^)
RDX	(100)	−101,929.85	−4428.33	54.57	4.630	4.284	90.000	19.83	2.75
	(010)	−102,290.94	−4444.02	38.88	4.284	5.273	90.000	22.59	1.72
	**(001)**	**−102,367.25**	**−4447.33**	**35.56**	**5.273**	**4.630**	**90.000**	**24.41**	**1.46**
	(110)	−101,806.89	−4422.99	59.91	4.284	7.017	90.000	30.06	1.99
	(101)	−101,784.11	−4422.00	60.90	6.794	4.630	90.000	31.45	1.94
	(011)	−101,940.71	−4428.80	54.09	5.273	6.307	90.000	33.26	1.63
	(111)	−101,543.49	−4411.54	71.35	6.793	6.307	64.645	38.72	1.84
	(210)	−101,373.64	−4404.16	78.73	4.284	10.655	90.000	45.64	1.72
HMX	(100)	−33,128.81	−1439.28	28.04	2.940	4.413	90.000	12.98	2.16
	(010)	−33,322.18	−1447.68	19.64	4.413	2.613	90.000	11.53	1.70
	(001)	−33,500.61	−1455.43	11.89	2.613	2.940	77.320	7.50	1.59
	(110)	−32,957.35	−1431.83	35.49	4.413	3.479	90.000	15.35	2.31
	(101)	−33,129.62	−1439.31	28.00	5.129	2.940	83.578	14.99	1.87
	**(011)**	**−33,342.86**	**−1448.58**	**18.74**	**2.613**	**5.303**	**83.009**	**13.75**	**1.36**
	(1 $\bar{1}$ 0)	−32,973.36	−1432.52	34.79	4.341	4.413	90.000	19.16	1.82
	(111)	−32,901.97	−1429.42	37.89	3.479	5.129	73.248	17.08	2.22
	(1$\bar{1}$$\bar{1}$)	−32,834.41	−1426.49	40.83	4.341	5.129	67.514	20.57	1.98
	(210)	−32,457.30	−1410.10	57.21	4.413	5.888	90.000	25.98	2.20

## Data Availability

The data that support the findings of this study are available from the corresponding author upon reasonable request.

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
