# Peer review of "Characterization of Nano-Scale Parallel Lamellar Defects in RDX and HMX Single Crystals by Two-Dimension Small Angle X-ray Scattering"

_molecules, 2022, doi:10.3390/molecules27123871_

Round 1

Reviewer 1 Report

The manuscript by Zhang reports the nanodefect study of explosive RDX and HMX single crystals. The work is performed on a high methodological level and very well presented. The only one critical note I would like to suggest to the authors, is that the introduction/review part should be appended by some more actual results reported in recent years. 

Reviewer 2 Report

1.    During the study, why authors select different crystal face slab for HMX and RDX?

2.   Under the subtitle 3.1, what about the nano-scale defect structure data of HMX single crystal such as average thicknes of nano-scale lamellar defect and average diameter.

3. I suggest to check the written way of orientation direction S12 in text and in figure 3 caption.

4.    In figure 4, is that SAXS data analyzing by the Ruland method for both RDX and HMX?  

5.   I sugest to check the crystal plane of lamellar defect in RDX and HMX, which I found different as a type error in 3.2 and 3.3 sections along with the conclusions section.

6.     I also suggest to cite the latest publications.

7. I suggest to check the Figure numbers in the manuscript.

Reviewer 3 Report

In this work, the nano-scale crystal defects of RDX and HMX single crystals are characterized by two-dimensional SAXS. The SAXS patterns from different incident directions are used to derive the shape and size of nano-scale defects. The XRD and first principles calculation are also adopted to interpret the crystallographical orientations and mechanisms. The results and analyses, although mostly qualitative, are interesting and solid. The manuscript can be accepted for publication after the following comments are properly handled. 

1. The manuscript should be carefully proofread for English, grammar errors and typos. The symbols should be italic; space is necessary between symobls and units; no intention before where after each equation.

2. How about radiation damage to the organic crystals during SAXS? How much time and radiation dose for each test?

3. How is the cut-off q selected for fitting in Fig. 3c? How does this cut-off affect the fitted Rg? The fitting errors should be added for rigorosity.

4. Discussions with the nano-CT characterizations on explosive crystals in previous literature are beneficial.
